# Coupling Influence between Recycled Ceramics and Grazed Hollow Beads on Mechanical Properties and Thermal Conductivity of Recycled Thermal Insulation Concrete

**DOI:** 10.3390/ma14164695

**Published:** 2021-08-20

**Authors:** Ying Yu, Ben Li, Dongmei Luo

**Affiliations:** Advanced and Sustainable Infrastructure Materials Group, School of Transportation, Civil Engineering and Architecture, Foshan University, Foshan 528000, China; dmluo@fosu.edu.cn

**Keywords:** recycled ceramics (RCE), grazed hollow beads (GHB), recycled thermal insulation concrete (RTIC), mechanical properties, thermal conductivity, materials properties

## Abstract

This paper investigated the influence of recycled ceramics and grazed hollow beads on the mechanical, thermal conductivity and material properties of concrete. The results showed that the concentration of recycled ceramics and grazed hollow beads has significant optimization on the workability and thermal properties of the concrete. However, the superabundant concentration can reduce the hydration degree of the concrete, which results in the suppressed production of C-S-H gel and the increase of material defects. In summary, considering the coordinated development of key factors such as thermal insulation properties, mechanical properties and microstructure, 10% RCE and 60% GHB are the optimal material system design methods.

## 1. Introduction

Environmental pollution and resource and energy shortages have become the main problems hindering the development of industrialization and urban construction [1,2,3,4,5]. With the continual improvement of industrialization processes, the accumulated storage volume of industrial waste has rapidly increased, causing irreversible environmental damage, such as soil erosion, air pollution, landscape destruction, deterioration of wildlife habitat and serious personal and property losses [6,7,8]. As the largest consumption carrier of industrial waste, concrete is gradually being recognized by government departments and the public for its role in the sustainable development of green ecology. Simultaneously, with the supersaturated population growth rate and the development of urbanization, the demand for concrete still maintains a relatively high volume. Recycling waste materials into concrete may be a necessary means for solving the above problems. On the one hand, it absorbs recyclable industrial waste to prepare new green building materials. On the other hand, it saves the non-renewable raw materials consumed in the traditional concrete preparation process [9,10,11,12,13,14,15,16]. As the world’s largest ceramic producer and consumer, Foshan had produced more than 9 million tons of ceramics with a scrap rate of 5–25% by 2020 [17,18,19,20]. However, China’s disposal of waste ceramics is still in the stage of centralized burial, which has largely caused a waste of resources and environmental pollution. The resource utilization methods and methods of waste ceramic disposal have become the hotspots and key points in the current research in the field. In recent years, many scholars have carried out research on recycled ceramic aggregate concrete. The research results can be summarized as: (1) Due to the effect of the characteristics of ceramic aggregates, the replacement of natural coarse aggregates with ceramic coarse aggregates will lead to a decrease in the relative volume and fluidity of the mortar in the concrete. (2) Simultaneously, the compressive strength decreases as the replacement rate of ceramic coarse aggregate increases. (3) The flexural strength of recycled ceramic coarse aggregate concrete decreases with the increase of the ceramic coarse aggregate replacement rate because the crushing index of ceramic coarse aggregate is higher than that of natural coarse aggregate [21,22,23,24,25].

Based on the characteristics that the mechanical properties of recycled ceramic coarse aggregate concrete are lower than ordinary concrete, recycled ceramic concrete is more widely used in the preparation of enclosure structures. The engineering application of the envelope structure with heat preservation and heat insulation properties can effectively alleviate the problem of energy shortage in China which conforms to the fundamental national policy of energy saving and emission reduction [26,27,28,29,30]. Based on ceramic coarse aggregates, lightweight thermal insulation concrete structures can be prepared. However, due to the complex material properties of ceramic coarse aggregates, thermal insulation concrete has the disadvantages of uneven temperature distribution and high thermal conductivity. At present, the preparation of thermal insulation concrete structures with ceramic coarse aggregates is still unable to fully meet engineering needs. How to improve the thermal insulation performance of recycled ceramic coarse aggregate concrete and meet the needs of engineering applications is still an urgent problem to be solved. In recent years, as a new type of lightweight aggregate, grazed hollow beads (GHB) have the characteristics of a porous inner surface and a glass-like outer surface, and have the function of a micro pump, which is easy to combine with cementitious materials. At the same time, the concrete prepared with GHB as lightweight aggregate has good mechanical and thermal insulation properties [16,31,32,33,34,35,36,37,38]. However, the research on the mechanical properties, thermal conductivity and material properties of cement-based material systems prepared by combining GHB and ceramic coarse aggregates is insufficient, and it is still necessary to enrich the mechanism research of the basic material system to support its engineering applications.

The goal of this study is to fill these research gaps by investigating the mechanical properties, thermal conductivity and material characteristics of recycled thermal insulation concrete (RTIC) that incorporates grazed hollow beads and recycled ceramics as replacement fine and coarse aggregates. Slump, density, splitting strength, compressive strength, axial compressive strength, elastic modulus and thermal conductivity were measured to evaluate the impact of grazed hollow beads and recycled ceramics on the macroperformance. Material morphology, hydration products and functional groups were investigated to determine the mesoscopic properties of recycled thermal insulation concrete. In summary, based on the above research results, new ideas are provided for the actual use of lightweight thermal insulation cement-based materials in civil engineering, such as the construction of thermal insulation structural floor slabs and the preparation of special concrete structures.

## 2. Materials and Methods

### 2.1. Raw Materials and Mixing Proportions

Ordinary Portland Cement (P.O. 42.5 N), local nature sand (NSD), grazed hollow beads (GHB), recycled ceramics (RCE) and nature stone (NSE) were used throughout the experiment. The mechanical properties of cement and chemical composition of cement are shown in Table 1 and Table 2. The particle size distribution curves of nature sand and nature stone are shown in Figure 1 and Figure 2. The physical properties of local nature sand, grazed hollow beads, recycled ceramics and nature stone are shown in Table 3. In this paper, the grazed hollow beads were used for replacing nature sand in different volume ratios as 0, 20, 40, 60% and the recycled ceramics were used for replacing nature stone in different mass ratios as 0, 10, 20, 30%. The replacement methods for grazed hollow beads and recycled ceramics are shown in Figure 3. The water-to-cement ratio (*w*/*c*) was 0.5 and the mixing proportions of the concrete mixtures are listed in Table 4. In the process of preparing concrete, the use of traditional mixing preparation methods caused the concrete slump to fail to meet the specification requirements due to the high water absorption of RCE and GHB. In order to solve this problem, the RCE was pre-wetted during the preparation process, and then the GHB was pretreated by the free water-cement ratio method, and the moisture required to be absorbed by the GHB was calculated. Finally, the actual water demand during concrete preparation was calculated for final mixing preparation.

### 2.2. Specimen Casting and Curing Conditions

The recycled thermal insulation concrete (RTIC) specimens had dimensions of 150 mm × 150 mm × 150 mm (144 pieces) and 300 mm × 300 mm × 30 mm (72 pieces) according to the Chinese standard GB/T 50081-2002 [39] and GB/T 10294-2008 [40]. All experimental specimens were demolded after 24 h and maintained for 28 days in a steam curing room with a temperature of 20 ± 2 °C and a relative humidity of 95%.

### 2.3. Experimental Methods

#### 2.3.1. Workability Properties Test

The workability of recycled thermal insulation concrete based on the coupling influence between GHB and RCE was determined and conducted through the slump and density test according to Chinese standard GB/T50080-2016 [41] and GB/T50080-2002 [42].

#### 2.3.2. Mechanical Properties Test

A total of 144 RTIC specimens (150 mm × 150 mm × 150 mm) were prepared for series experiments of compressive strength (72 pieces) and splitting strength (72 pieces) according to Chinese standard GB/T50081-2002 [39].

#### 2.3.3. Thermal Insulation Properties Test

The thermal insulation property of concrete is usually reflected by thermal conductivity. In this paper, a total of 72 specimens (300 mm × 300 mm × 30 mm) were prepared to determine the thermal conductivity of concrete specimens based on the steady-state and double-plate method (shown in Figure 4).

#### 2.3.4. Materials Characterization Tests

To further explore the effect of RCE and GHB on the mechanical properties and thermal conductivity early performance of recycled thermal insulation concrete, several macro-experimental methods were conducted. A total of 80 experimental samples were subjected to scanning electron microscopy (Hitachi, Shenzhen, China) (SEM, 33 samples), X-ray diffraction (Haoyuan, Dandong, China) (XRD, 24 samples), and Fourier-transform infrared spectroscopy (BRUKER, Shanghai, China) (FTIR, 24 samples) to analyze the changes in the micromorphology, hydration products, and chemical bonding or molecules. The preparation and drying conditions of these samples were following Chinese standard GB/T 16594-2008 [43], GB/T 30904-2014 [44], and ISO 19618-2017 [45]. In summary, the experimental flowcharts in this paper are shown in Figure 5.

## 3. Results and Discussion

### 3.1. Coupled Influence of RCE and GHB on Macroscopic Comprehensive Characteristics of RTIC

In this section, the macroscopic comprehensive performance of RTIC is studied. This section mainly explores the influence of GHB and RCE on the working performance, lightweight performance, mechanical properties and thermal conductivity of RTIC under the coupling effects of different doping amounts.

#### 3.1.1. Workability and Density Properties of RTIC

The experimental results and changes of slump subjected to the coupling influence between GHB and RCE on fresh concrete are shown in Figure 6a,b. The results showed that, (1) when the content of RCE was constant, the slump degree of concrete increased with the increase of GHB content. Compared with RCE0GHB0 (45 mm), RCE0GHB20 (80 mm), RCE0GHB40 (110 mm) and RCE0GHB60 (115 mm) increased by 77.8, 144.4 and 115.6%, respectively. (2) However, when the content of GHB was constant, the influence of RCE on the slump properties of concrete was not obvious. Compared with the reference concrete, the change of collapse degree was 11.1, 1.0 and −11.1% within the content range of RCE from 10 to 30%. (3) Similarly, the combined effect of GHB and RCE could better improve the workability performance of RTIC. When the blending contents were RCE0GHB60 (115 mm) and RCE30GHB60 (115 mm), the material system could obtain the best slump value. (4) The improved workability performance of RTIC was mainly attributed to the influence of GHB. This is because GHB has good water retention properties and can reduce the friction between the aggregates, which can eventually lead to provide the required water phase during the material preparation process. Nevertheless, RCE also has good water absorption, it has little effect on the workability of concrete. This is mainly because RCE needs to be pre-wetted in advance to reach saturation, which reduces the effect of improving flow performance. Figure 7a,b show the experimental results and changes of density subject to the coupling influence between GHB and RCE on concrete. The results show that: (1) compared with RCE0GHB0, the density of the concrete with the increasing single incorporation of RCE (10~30%) and GHB (20~60%) decreased by 1.6, 3.7, 5.3 and 0.8, 1.6, 2.5%, respectively. (2) RCE has a significant effect on reducing the density of the concrete. Although the density of GHB is much lower than that of natural sand, it absorbs a lot of water during the concrete preparation process, which is not obvious for the reduction of concrete density. (3) In the compound incorporation of GHB and RCE, the negative density change of RCE10GHB20 was the smallest, which had the best combined effect on reducing the density of concrete.

#### 3.1.2. Analysis of the Compressive Strength of RTIC

The experimental results and changes of compressive strength are shown in Figure 8a,b. It can be seen that, (1) GHB had a decisive influence on the compressive strength of concrete. Compared with RCE0GHB0, the compressive strength of RCE0GHB20, RCE0GHB40 and RCE0GHB60 decreased by 5.7, 15.6 and 22.7%, respectively. With the increase of GHB content, the compressive strength of concrete was further reduced. That is because GHB is a brittle material, which is prone to brittle fracture and decomposition due to stress concentration under the action of load, thereby forming more pore defects in the concrete [46,47]. In summary, GHB has no significant contribution to the compressive strength development of concrete. (2) Furthermore, incorporating ceramic waste aggregate as a partial replacement of natural coarse aggregate also reduced the compressive strength. The compressive strength reduction of RCE10GHB0, RCE20GHB0 and RCE30GHB0 was 5.9, 11.1 and 13.3% compared with RCE0GHB0. The concrete prepared with RCE was prone to through cracks due to aggregate damage in the process of compressive load failure due to low crush value and porous brittle material properties [7,48,49]. (3) The coupling of GHB and RCE had no positive effect on the compressive strength of RTIC. Comprehensive comparison of the change in compressive strength, RCE30GHB60 had the largest decrease in compressive strength, while RCE10GHB20 had the smallest decrease.

#### 3.1.3. Analysis on Splitting Tensile Strength of RTIC

Figure 9a,b shows the experimental results and changes of splitting tensile strength of RTIC. Splitting tensile strength decreased significantly with the increase of the incorporation of GHB and RCE. In the process of splitting failure, the GHB and RCE in the concrete deteriorated (even powdery failure) before the natural aggregates. At the same time, the pore structure and micro cracks in the concrete will increase and form more mechanically weak areas, which will eventually lead to a negative impact on the splitting performance of concrete based on the coupling influence between GHB and RCE. In addition, RCE30GHB60 had the largest reduction in split tensile strength (−1.06 MPa), while RCE10GHB20 had the smallest reduction in split tensile strength (−0.68 MPa), which is the best ratio to ensure split tensile performance.

#### 3.1.4. Insulation Properties of RTIC

The experimental results and changes of thermal conductivity and insulation property values of RTIC subject to the coupling influence between GHB and RCE are shown in Figure 10a–c. The results show that, (1) the thermal conductivity decreased as the content of GHB increased. When the GHB content was 60%, RTIC had lower thermal conductivity and performance. The significant decrease in thermal conductivity was mainly attributed to the good thermal insulation properties of GHB. When GHB is distributed in concrete, the pore structure and internal spaces are filled with foam material, which hinders the conduction and dispersion of heat, thereby reducing the thermal conductivity of the concrete. (2) However, the thermal conductivity decreased slightly with the increase of RCE, but when RCE and GHB were coupled, the thermal conductivity did not change significantly. This is because the thermal conductivity of ceramics is not significantly different from that of natural coarse aggregates. Compared with natural coarse aggregate, ceramic coarse aggregate has more microcracks, and the porosity of ceramic coarse aggregate is greater than that of ordinary concrete. These factors have a limited influence on reducing the thermal conductivity of concrete. (3) In summary, GHB played a decisive role in reducing the thermal conductivity and improving the thermal insulation performance of RTIC. Considering the mechanical properties and thermal insulation properties of concrete, the mixing amount of RCE is 10–20%, and the mixing amount of GHB is 40–60%.

### 3.2. Coupled Influence of RCE and GHB on Material Properties of RTIC

In order to further explore the modification mechanism of RTIC on the comprehensive properties of concrete, a set of meso-experimental methods to explore the coupled effects of GHB and RCE on RTIC’s microstructure, material morphology and key hydration products are discussed in this section.

#### 3.2.1. Material Morphology Analysis of RTIC

The analysis of the material morphology of RTIC is shown in Figure 11. Compared with RCE0GHB0, RCE affected the cured form of C-S-H gel in concrete. As the content of RCE increased, more flocculent C-S-H gels were formed and attached to the surface of the unhydrated cement stone. At the same time, RCE affected the internal spatial distribution of the concrete and caused more microcracks or an increase in obvious pores, which explained why the mechanical properties of the concrete decreased due to the RCE aggregate. In addition, GHB caused unhydrated cement particles to adhere to the surface of C-S-H. With the increase in the amount of GHB, the volume and pore size of the pores in the cement stone increased significantly, thereby improving the thermal insulation properties of the concrete, but impairing the mechanical properties.

#### 3.2.2. Hydration Products Analysis of RTIC

The experimental results of the crystal composition and hydration products of RTIC are shown in Figure 12. The results show that, (1) with the increase of RCE and GHB, the characteristic peak intensity of hydration products CH and C-S-H decreased. The increase of RCE and GHB inhibited the formation of the main hydration products in the concrete, which was not conducive to the development of mechanical properties. (2) RCE and GHB had a significant effect on the formation of CaCO_3_. The increase of CaCO_3_ leads to premature calcification or aging of concrete, which is very unfavorable to the development of mechanical properties. (3) Further analysis found that GHB had a promoting effect on the formation of Tobermorite. With the increase of GHB, the characteristic peak intensity and crystal yield of Tobermorite increased. Tobermorite is a kind of hydrated calcium silicate with high crystallinity. It has strong thermal insulation performance and can improve the thermal insulation performance of concrete. Therefore, with the increase of GHB, the thermal insulation performance of concrete is also enhanced.

#### 3.2.3. Functional Group Changes of RCTIC

The effect of RCE and GHB on the functional groups of concrete hydration products is shown in Figure 13. The shrinkage vibration of H-O-H (3400) functional group and the bending vibration of H-O-H (1640) significantly changed with the increasing content of RCE and GHB. H-O-H (3400) functional group had a more pronounced oscillation range, because RCE and GHB have high water absorption and can store a large amount of water phase. At the same time, the shrinkage vibration of the -OH (3637) functional group in CH increased significantly with the content of RCE and GHB based on the influence of CaCO_3_ production in concrete.

### 3.3. Mathematical Representation of the Coupled Influence of RCE and GHB

The effects of RCE and GHB on RTIC workability, compressive strength, splitting tensile strength and insulation properties can be quantitatively determined by the correlation principle. The interaction between these two parameters is determined as follows:(1)$$k_{1}=\frac{Slump[Combined\;RCE\;and\;GHB\;Content]}{Slump[Control\;Group]}$$
(2)$$k_{2}=\frac{Compressive\;strength[Combined\;RCE\;and\;GHB\;Content]}{Compressive\;strength[Control\;Group]}$$
(3)$$k_{3}=\frac{Splitting\;tensile\;strength[Combined\;RCE\;and\;GHB\;Content]}{Splitting\;tensile\;strength[Control\;Group]}$$
(4)$$k_{4}=\frac{Insulation\;properties[Combined\;RCE\;and\;GHB\;Content]}{Insulation\;properties[Control\;Group]}$$
(5)$$K={ℜ}_{1}k_{1}+{ℜ}_{2}k_{2}+{ℜ}_{3}k_{3}+{ℜ}_{4}k_{4}$$
where $k_{1}$ is the coefficient of the influence on liquidity, $k_{2}$ is the coefficient of influence on compressive strength, $k_{3}$ is the coefficient of influence on splitting tensile strength, $k_{4}$ is the coefficient of influence on insulation, ℜ is the correlation parameter between RCE and GHB content, *K* is the quantitative characterization of the coupled effects of RCE and GHB content on concrete comprehensive characteristics.

The calculation results of the influence coefficient of RCE and GHB on concrete performance are shown in Figure 14. The following conclusions can be reached. (1) Considering the mechanical properties and thermal insulation properties of concrete, the best content of RTIC is RCE10GHB60. (2) The allowable content of GHB used for general engineering needs can be relaxed in order to improve thermal insulation performance while ensuring the quality of the concrete. (3) The content of RCE needs to be strictly controlled, and its reasonable replacement rate should not exceed 20% of the natural coarse aggregate.

## 4. Conclusions

This paper investigated the influence of RCE and GHB on the mechanical, thermal conductivity and material properties of concrete, and the following conclusions can be drawn:RCE can inhibit the density and mechanical properties of RTIC, but it has little effect on the working and thermal insulation properties of RTIC. GHB can improve the workability and insulation properties of RTIC (155.6 and 91.9% maximum increase), but decreases the density and mechanical properties (9.9 and 33.9% maximum decrease).The incorporation of GHB and RCE inhibited the formation of CH and C-S-H and promoted the increase of CaCO_3_, which was detrimental to the macroscopic mechanical properties of RTIC. Meanwhile, the increase in GHB led to an increase in the production of Tobermorite, which largely improves the insulation properties of RTIC.GHB and RCE can be used to prepare regenerated thermal insulation concrete due to the synergistic development of mechanical properties and thermal insulation properties. It is considered that 10% RCE and 60% GHB are the best doping amounts. The new thermal insulation concrete material prepared based on this material design system can be used in practical projects such as prefabricated roof structures in civil engineering, special transportation pipelines, and link bridges in equipment and accessory buildings.GHB and RCE can be used to prepare regenerated thermal insulation concrete due to the synergistic development of mechanical properties and thermal insulation properties. It is considered that 10% RCE and 60% GHB are the best doping amounts.

## Figures and Tables

**Figure 1 materials-14-04695-f001:**
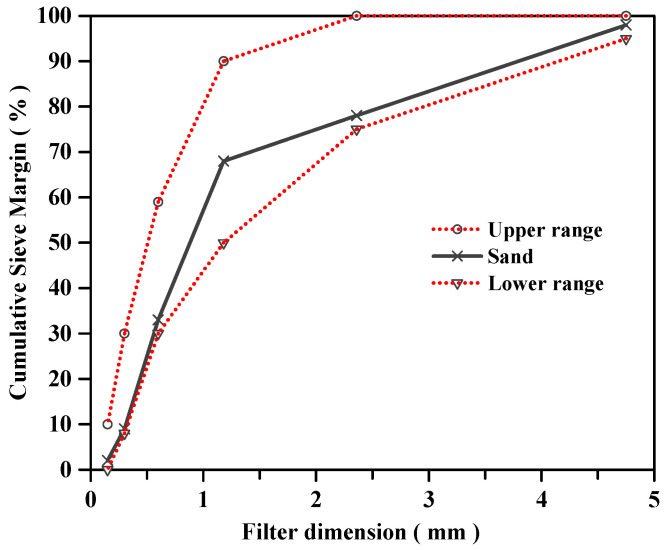
The particle size distribution curves of nature sand.

**Figure 2 materials-14-04695-f002:**
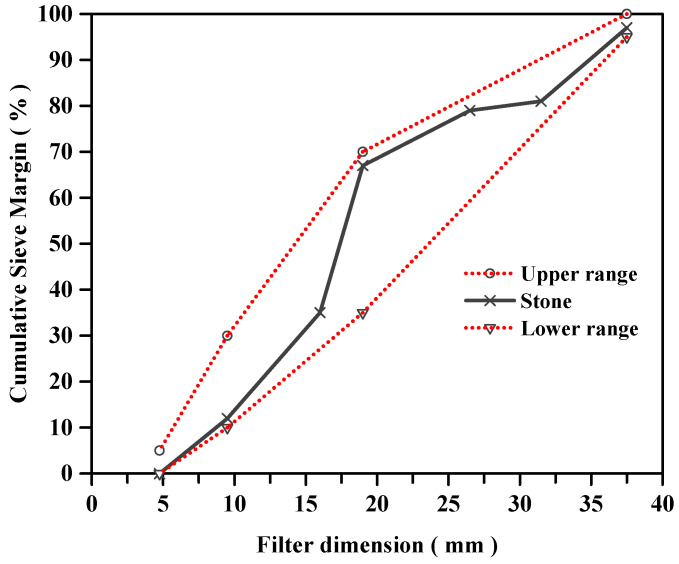
The particle size distribution curves of nature stone.

**Figure 3 materials-14-04695-f003:**
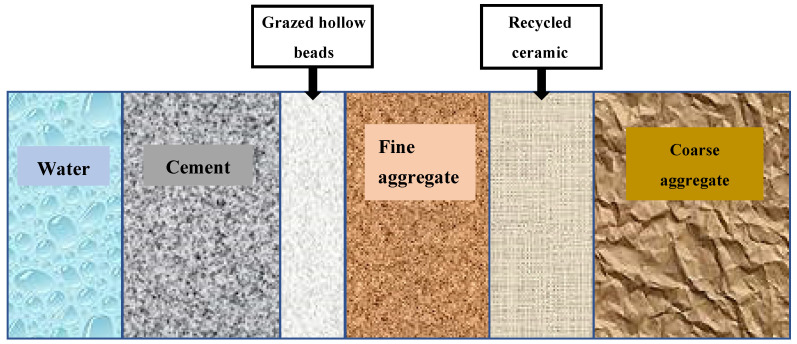
The replacement methods for RCE and GHB.

**Figure 4 materials-14-04695-f004:**
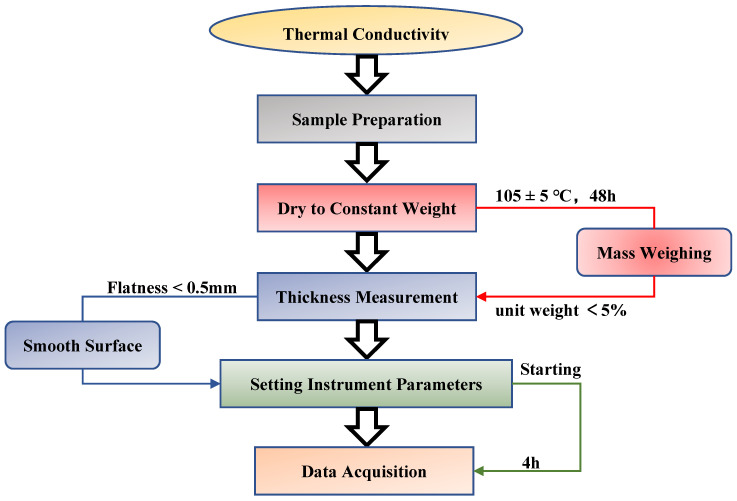
Measurement of thermal conductivity on the RTIC.

**Figure 5 materials-14-04695-f005:**
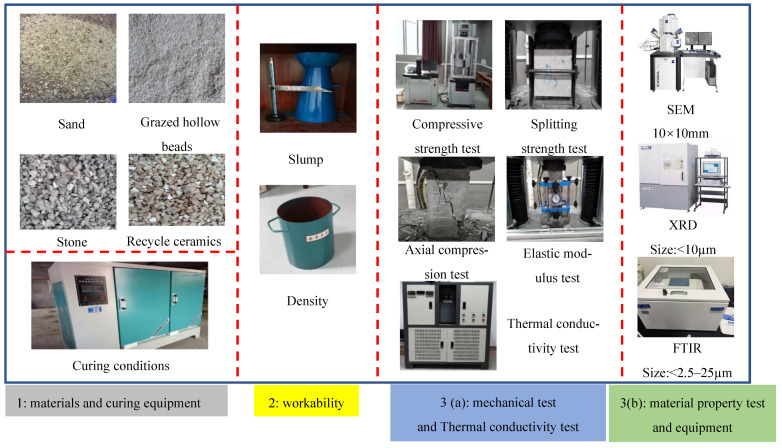
Experimental flowchart.

**Figure 6 materials-14-04695-f006:**
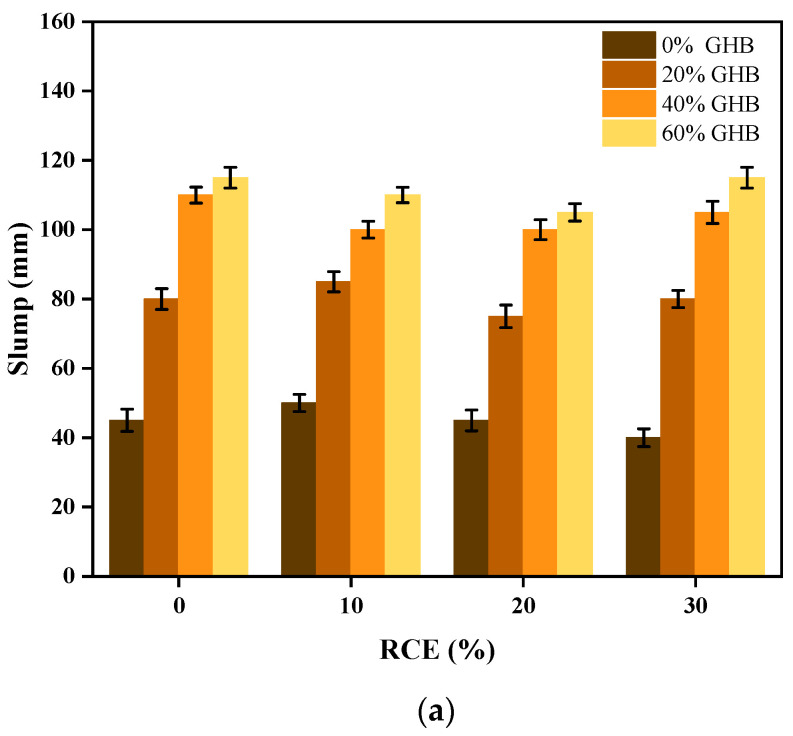
The experimental results and changes of slump subjected to the coupling influence between GHB and RCE on fresh concrete. (**a**) The experimental results of slump on fresh concrete; (**b**) the changes of slump on fresh concrete.

**Figure 7 materials-14-04695-f007:**
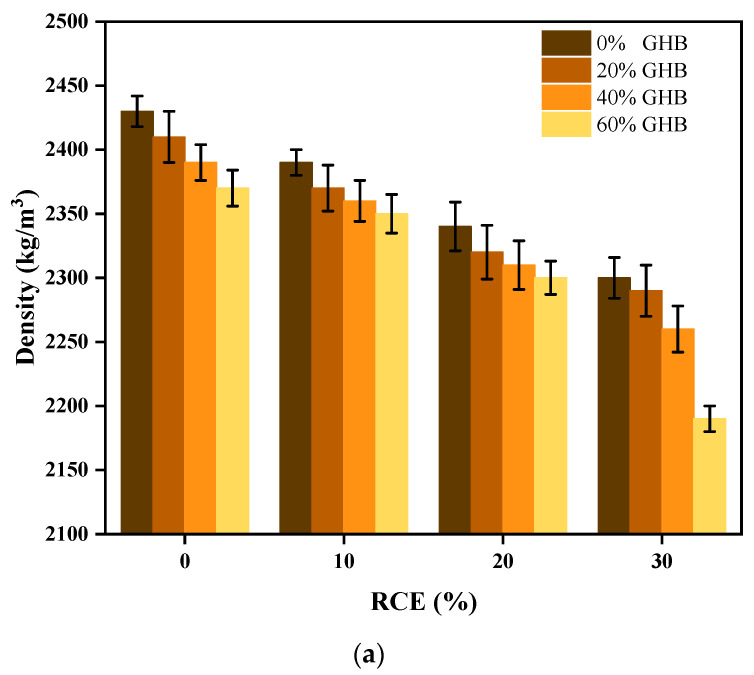
The experimental results and changes of density subject to the coupling influence between GHB and RCE on RTIC. (**a**) The experimental results of density on concrete; (**b**) the changes of density on fresh concrete.

**Figure 8 materials-14-04695-f008:**
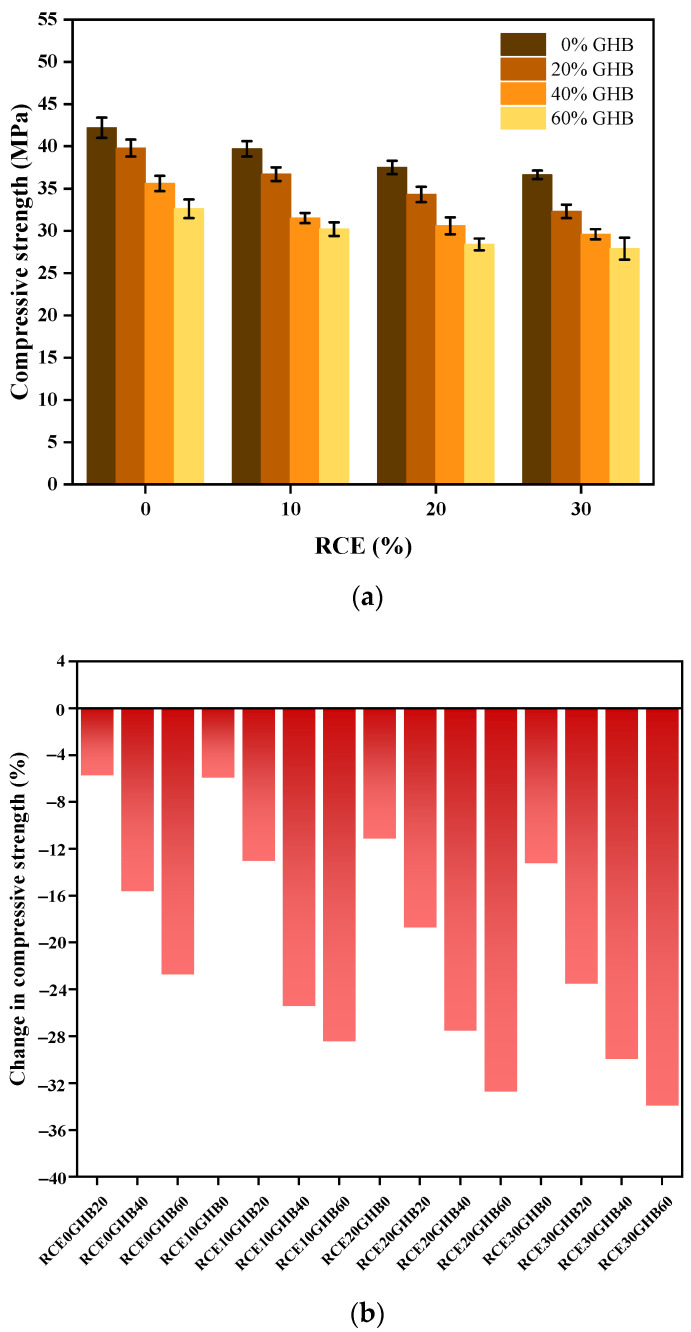
The experimental results and changes of compressive strength subject to the coupling influence between GHB and RCE on RTIC. (**a**) The experimental results of compressive strength of RTIC; (**b**) the changes of compressive strength of RTIC.

**Figure 9 materials-14-04695-f009:**
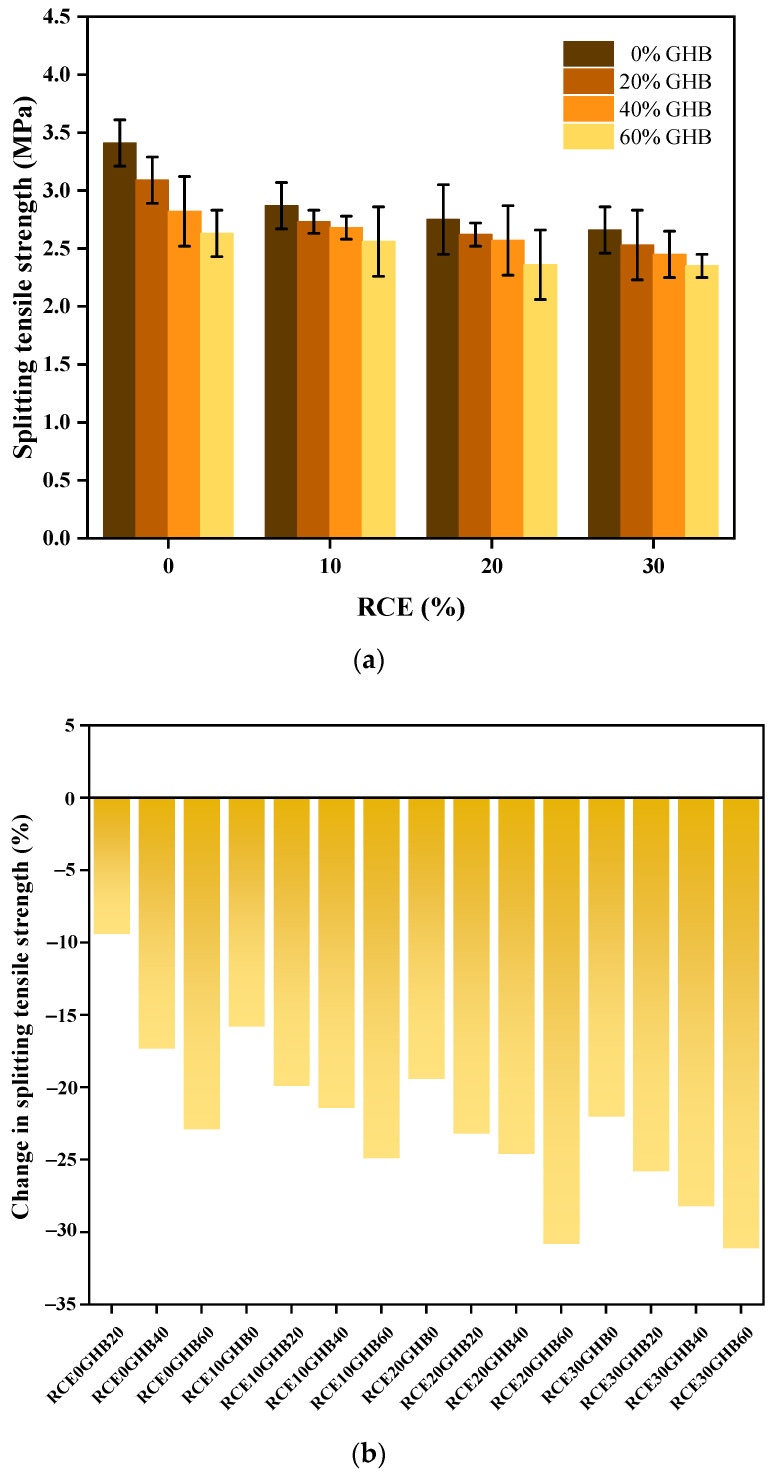
The experimental results and changes of splitting tensile strength subject to the coupling influence between GHB and RCE on RTIC. (**a**) The experimental results of splitting tensile strength of RTIC; (**b**) the changes of splitting tensile strength of RTIC.

**Figure 10 materials-14-04695-f010:**
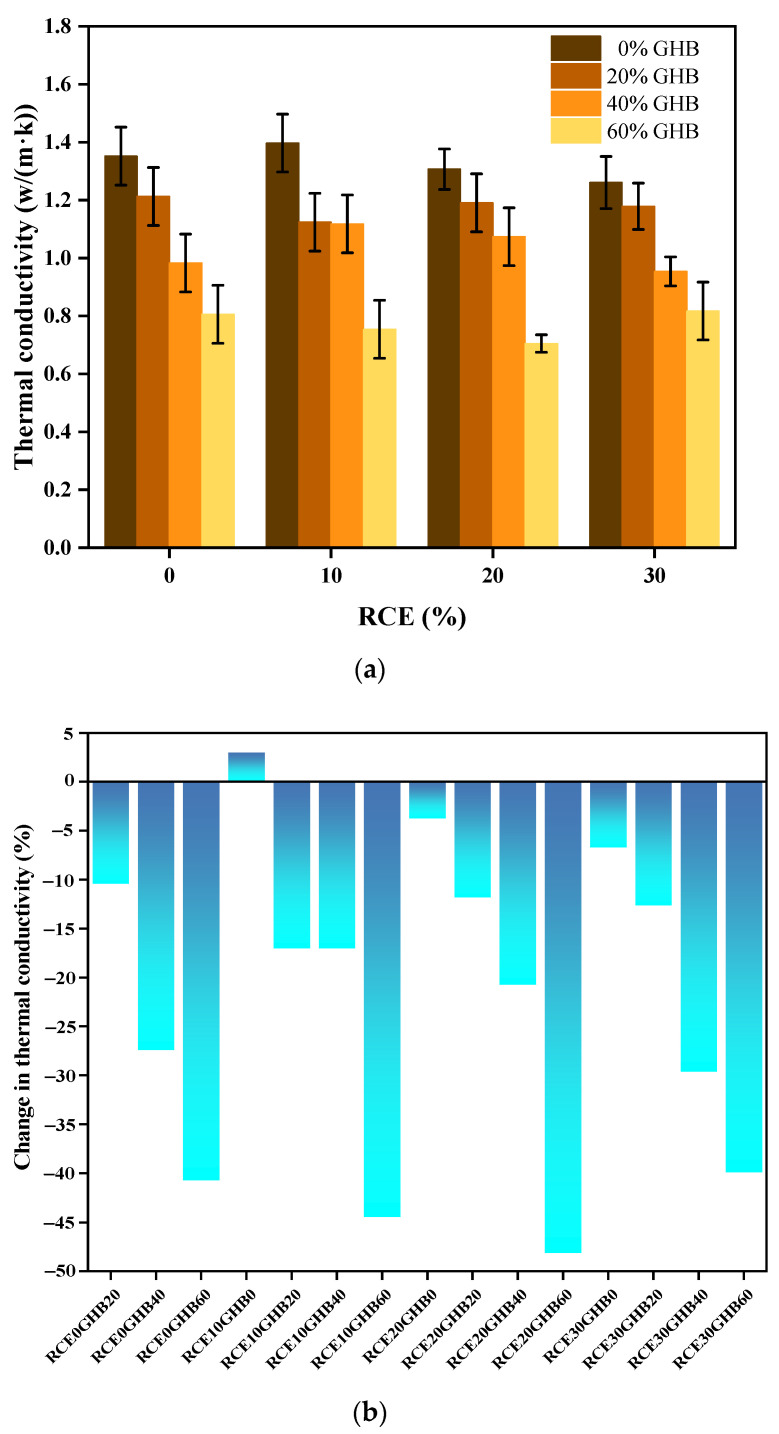
The experimental results and changes of thermal conductivity and insulation property values of RTIC subject to the coupling influence between GHB and RCE. (**a**) The experimental results of thermal conductivity of RTIC; (**b**) the changes of thermal conductivity of RTIC. (**c**) Insulation property values of RTIC.

**Figure 11 materials-14-04695-f011:**
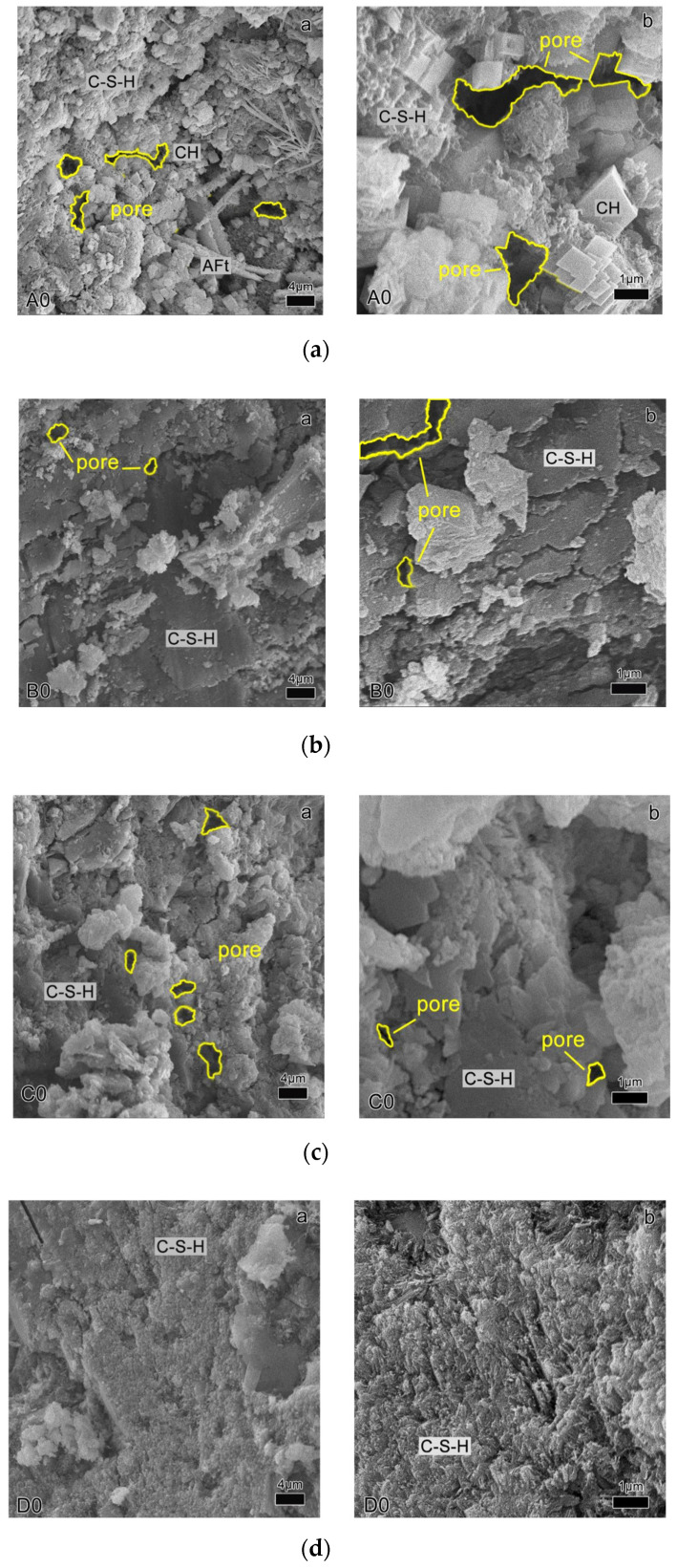
The analysis of the material morphology of RTIC based on SEM (2.0 k to 10.0 k). (**a**) RCE0GHB0; (**b**) RCE10GHB0; (**c**) RCE20GHB0; (**d**) RCE30GHB0; (**e**) RCE10GHB20; (**f**) RCE10GHB40; (**g**) RCE10GHB60.

**Figure 12 materials-14-04695-f012:**
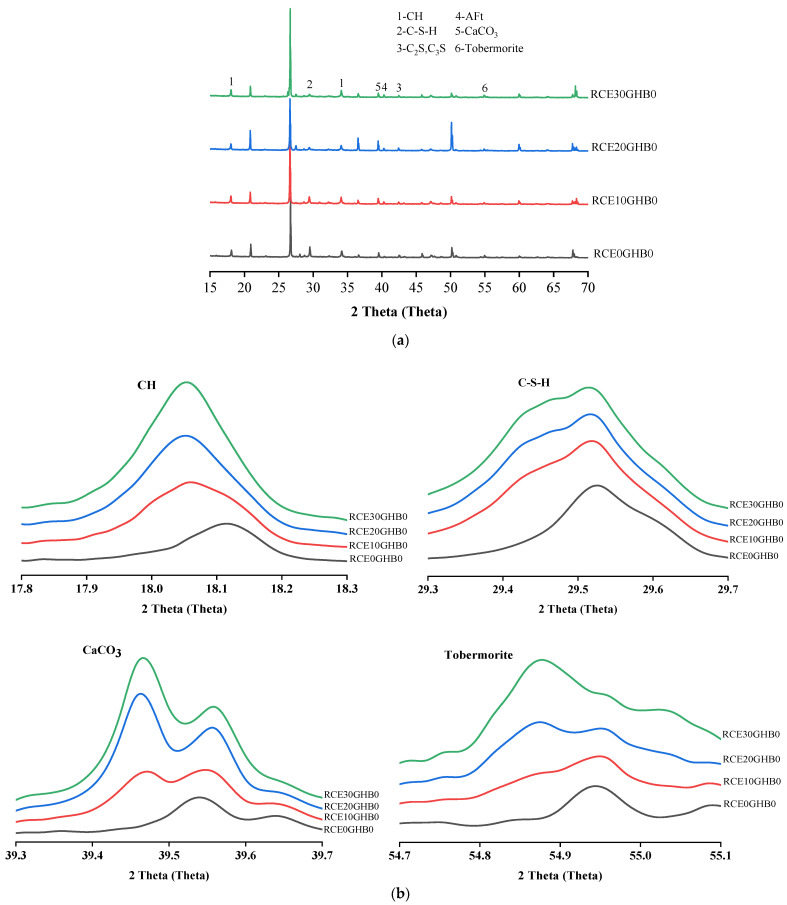
The experimental results of the crystal composition and hydration products of RTIC based on XRD. (**a**) The crystal composition and hydration products of RTIC (RCE0GHB0, RCE10GHB0, RCE20GHB0, RCE30GHB0); (**b**) CH, C-S-H, CaCO_3_, Tobermorite (RCE0GHB0, RCE10GHB0, RCE20GHB0, RCE30GHB0); (**c**) the crystal composition and hydration products of RTIC (RCE10GHB0, RCE10GHB20, RCE10GHB40, RCE10GHB60); (**d**) CH, C-S-H, CaCO_3_, Tobermorite (RCE10GHB0, RCE10GHB20, RCE10GHB40, RCE10GHB60).

**Figure 13 materials-14-04695-f013:**
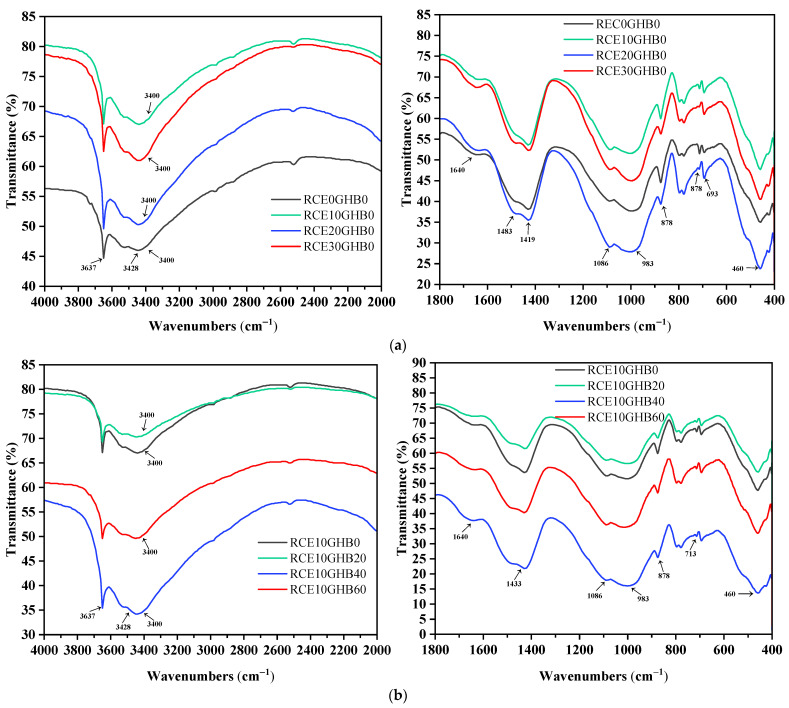
The effect of RCE and GHB on the functional groups of concrete hydration products. (**a**) The functional groups of RTIC (RCE0GHB0, RCE10GHB0, RCE20GHB0, RCE30GHB0); (**b**) the functional groups of RTIC (RCE10GHB0, RCE10GHB20, RCE10GHB40, RCE10GHB60).

**Figure 14 materials-14-04695-f014:**
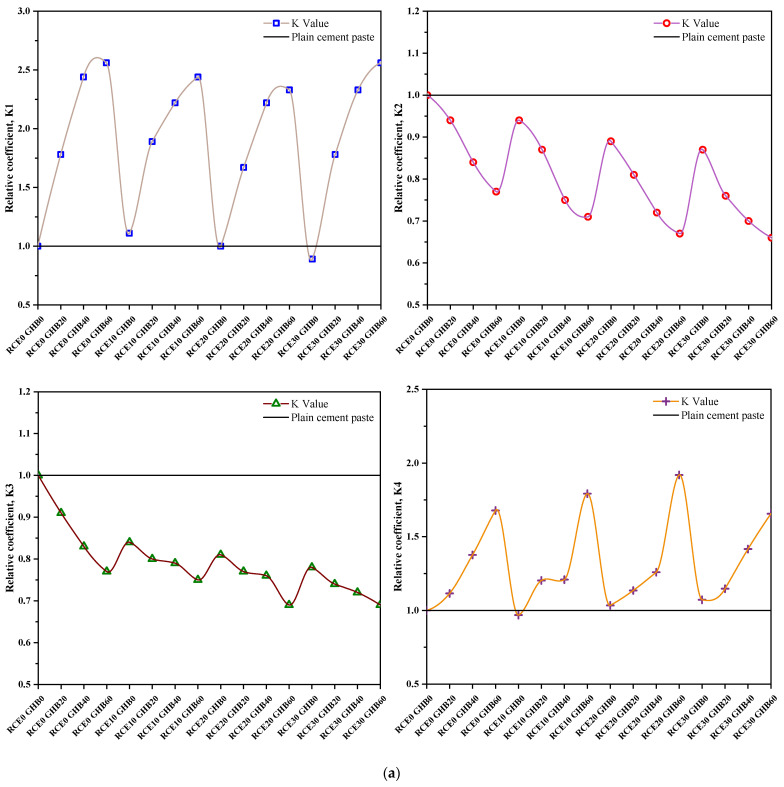
The calculation results of the influence coefficient of RCE and GHB on concrete performance. (**a**) K1, K2, K3, K4. (**b**) K.

**Table 1 materials-14-04695-t001:** Mechanical properties of Portland cement (MPa).

Flexural Strength (MPa)		Compressive Strength (MPa)		Fineness	Setting Time (min)	
3 days	28 days	3 days	28 days	1.2	Initial setting	Final setting
4.2 ± 0.2	7.5 ± 0.5	23.5 ± 0.8	43.2 ± 0.4		186	252

**Table 2 materials-14-04695-t002:** Chemical properties of Portland cement (%).

Compounds	SiO_2_	Al_2_O_3_	Fe_2_O_3_	CaO	MgO	SO_3_	Loss
Content	22.53	4.42	2.06	61.71	4.55	2.23	2.86

**Table 3 materials-14-04695-t003:** Properties of nature sand, grazed hollow beads, nature stone and recycled ceramic.

Physical Properties	Fine Aggregate		Coarse Aggregate	
Type	NSD	GHB	NSE	RCE
Accumulation density (kg/m^3^)	2050	99.52	1389	972
Apparent density (kg/m^3^)	2512	174.7	3045	1819
Water content (%)	0.81	0.5	0.12	2.51
Mass water absorption (%)	7.58	246	0.16	15.47
Crushing value (%)	-	-	5.32	15.95
Thermal conductivity (W/m·K)	-	0.072	-	-

Note: The crushing value is the ultimate compressive strength of the coarse aggregate used in the concrete preparation process. It is one of the required attributes in raw material performance research.

**Table 4 materials-14-04695-t004:** Mixing proportions of concrete mixtures (kg/m^3^).

W/C = 0.5	Cement	Water	NSD	NSE	RCE	GHB	SP
RCE0 GHB0	370	185	628	1218	0	0	1.9
RCE0 GHB20	370	202.8	628	1218	0	19.9	1.9
RCE0 GHB40	370	220.5	628	1218	0	39.8	1.9
RCE0 GHB60	370	238.3	628	1218	0	59.7	1.9
RCE10 GHB0	370	185	628	1096	122	0	1.9
RCE10 GHB20	370	202.8	628	1096	122	19.9	1.9
RCE10 GHB40	370	220.5	628	1096	122	39.8	1.9
RCE10 GHB60	370	238.3	628	1096	122	59.7	1.9
RCE20 GHB0	370	185	628	974	244	0	1.9
RCE20 GHB20	370	202.8	628	974	244	19.9	1.9
RCE20 GHB40	370	220.5	628	974	244	39.8	1.9
RCE20 GHB60	370	238.3	628	974	244	59.7	1.9
RCE30 GHB0	370	185	628	853	365	0	1.9
RCE30 GHB20	370	202.8	628	853	365	19.9	1.9
RCE30 GHB40	370	220.5	628	853	365	39.8	1.9
RCE30 GHB60	370	238.3	628	853	365	59.7	1.9

Note: SP is the high-efficiency polycarboxylic acid water reducer.

## Data Availability

Data available on request due to restrictions eg privacy or ethical.

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
