# Peer review of "Coupling Influence between Recycled Ceramics and Grazed Hollow Beads on Mechanical Properties and Thermal Conductivity of Recycled Thermal Insulation Concrete"

_materials, 2021, doi:10.3390/ma14164695_

Round 1
Reviewer 1 Report
The paper entitled “Coupling influence between recycled ceramics and grazed hollow beads on mechanical properties and thermal conductivity of recycled thermal insulation concrete”, investigates the influence of recycled ceramics and grazed hollow beads on the mechanical, thermal conductivity and materials properties of concrete. The results showed that the concentration of recycled ceramics and grazed hollow beads has significant optimization on workability and thermal properties of concrete.
COMMENTS
The Abstract should be better written to reflect the contents of the paper. The phrase “How to reasonably optimize the design of this material system to achieve the coordinated development of mechanical properties and thermal conductivity is one of the problems to be solved in the future”, could be written in the discussion and/or conclusions of the article.
The Figures and Tables should be embedded in the text and not be concentrated at the end of the paper.
The size of the text and numbers in the Figures must readable.
In the axes pf Figures, a space is needed before each left parenthesis “ (“
Line 271 and MDPI fig. after it, should be deleted.
Reviewer 2 Report
The article discuss the topic of the Coupling influence between recycled ceramics and grazed hollow beads on mechanical properties
and thermal conductivity of recycled thermal insulation concrete. The article presents valuable content. In my opinion article should be improved before potential publication. The following modification should be considered:
1. The abstract should contain major results obtained in the research (containing received values).
2. It is worth to add graphical abstract.
3. In the introduction part, a wider problem of utilization of many types of wastes in concrete structures should be considered and analysed,
especialy influencing the mechanical properties and thermal conductivity of concrete. Some literatures could be helpful: https://doi.org/10.1016/j.conbuildmat.2019.117794, https://doi.org/10.1016/j.wasman.2006.07.026, https://doi.org/10.3390/ma13194358.
4. I suggest to add separated point - Research significance - Please describe here the main essence of the research.
What was the inspiration for such an analysis? Why presented studies are so important?
5. Point 3.2. Analysis on the compressive strength of RTIC. What was the standard deviation of each series in case of compressive strength?
6. Line 270: 'All figures and tables should be cited in the main text as Figure 1, Table 1, etc.' Please remove it. Please
check the whole article if it contain similar bugs. For example please remove the logo of MDPI in line 271 etc.
7. In case of table 1 please add standard deviation for each value: for Flexural strength (MPa) and Compressive strength (MPa).
8. Table 3 - what is the Crushing value? Please explain in the text.
9. Table 4 - please explain abbreviations RC and SP.
10. The conclusion part could be more accurate.
11. It is recommended to indicate potential application of research results in civil engineering or another discipline.
Reviewer 3 Report
Dear Editor,
The topic of the paper is interesting and suits the Journal of MDPI Materials. However, a major revision is required before this manuscript is qualified to be published in this prestigious journal. The manuscript is needed to be revised grammatically. The authors are required to check the whole manuscript with a grammar specialist as it has several grammatical errors. Only after revising the manuscript based on the comments, the paper is suggested to be published in MDPI. Further information on various issues identified in the manuscript appears below:
- The introduction section needs to be revised. A paragraph should be dedicated to the importance of your work.
- The authors have done a great job on the literature review. However, the introduction needs more attention. More information on new materials related to the topic of this paper can be found here:
"Fracture Properties Evaluation of Cellulose Nanocrystals Cement Paste." Materials 13, no. 11 (2020): 2507.
- Please provide more detailed reasoning behind the behavior. The details should include rigid numbers or percentages.
- Please indicate how many samples for each experiment have been used. Please revise the other experiments respectively.
- Please add error bars to all figures.
- Please describe the process of each experiment. Also indicate the model of each tool that is used in the experiment. What is the accuracy of each machine? Please explain them accurately.
- Conclusion needs more elaboration. Please use more sentences containing percentages and illustrate the main conclusions in the manuscript. Please paraphrase your results and discussions and use them in the conclusion part.
Round 2
Reviewer 1 Report
The revised paper can now be accepted for publication.
Reviewer 2 Report
All remarks have been considered by authors.
The authors responded to comments of the reviewer precisely.
I suggest that article could be published.
Reviewer 3 Report
Dear Editor,
The authors have applied the comments and the paper is ready to be published.